# Cyclooxygenase-2 inhibition reduces stress-induced affective pathology

Joyonna Carrie Gamble-George[1,2], Rita Baldi[1], Lindsay Halladay[3], Adrina Kocharian[3], Nolan Hartley[1,2], Carolyn Grace Silva[1], Holly Roberts[1], Andre Haymer[1], Lawrence J Marnett[4], Andrew Holmes[3], Sachin Patel[1,2,5,6]*

[1]Department of Psychiatry and Behavioral Sciences, Vanderbilt University Medical Center, Nashville, United States; [2]Vanderbilt Brain Institute, Vanderbilt University, Nashville, United States; [3]Laboratory of Behavioral and Genomic Neuroscience, National Institute on Alcoholism and Alcohol Abuse, Bethesda, United States; [4]A.B. Hancock Jr. Memorial Laboratory for Cancer Research, Departments of Biochemistry, Chemistry, and Pharmacology, Vanderbilt Institute of Chemical Biology, Center in Molecular Toxicology, and Vanderbilt-Ingram Cancer Center, Nashville, United States; [5]Department of Molecular Physiology and Biophysics, Vanderbilt University School of Medicine, Nashville, United States; [6]Vanderbilt Kennedy Center for Human Development, Vanderbilt University Medical Center, Nashville, United States

**Abstract** Mood and anxiety disorders are the most prevalent psychiatric conditions and are exacerbated by stress. Recent studies have suggested cyclooxygenase-2 (COX-2) inhibition could represent a novel treatment approach or augmentation strategy for affective disorders including anxiety disorders and major depression. We show that traditional COX-2 inhibitors and a newly developed substrate-selective COX-2 inhibitor (SSCI) reduce a variety of stress-induced behavioral pathologies in mice. We found that these behavioral effects were associated with a dampening of neuronal excitability in the basolateral amygdala (BLA) ex vivo and in vivo, and were mediated by small-conductance calcium-activated potassium (SK) channel and CB1 cannabinoid receptor activation. Taken together, these data provide further support for the potential utility of SSCIs, as well as traditional COX-2 inhibitors, as novel treatment approaches for stress-related psychiatric disorders.

*For correspondence: sachin.patel@vanderbilt.edu

## Introduction

Mood and anxiety disorders including major depression, generalized anxiety disorder and posttraumatic stress disorder are the most common psychiatric disorders (*Kessler et al., 2005*; *2012*). Unfortunately, current pharmacological therapies, which act primarily via monoamine reuptake mechanisms, are only partially effective (*Griebel and Holmes, 2013*). Recent studies have begun to emphasize novel non-monoaminergic pathophysiological mechanisms underlying mood and anxiety disorders and support immunological (*Miller et al., 2009*; *Raison and Miller, 2013*), glutamatergic (*Abdallah et al., 2015*; *Dutta et al., 2015*), and bioactive lipid-based pharmacological approaches for treatment-resistant affective disorders (*Papakostas and Ionescu, 2015*). In this regard, there has been growing interest in cyclooxygenase-2 (COX-2) as a bioactive lipid-mechanism-based target for the treatment of affective disorders (*Akhondzadeh and Jafari, 2010*; *Fond et al., 2014*; *Hermanson et al., 2014*).

**eLife digest** People may experience a stressful or traumatic event that can cause anxiety in their day-to-day lives. For some people, this anxiety can persist and develop into a disorder, such as posttraumatic stress disorder (PTSD) or acute stress disorder. Millions of people worldwide suffer from these disorders, but there are few classes of drug treatments currently available. Some preliminary studies in mice have suggested that anti-inflammatory drugs called cyclooxygenase-2 (COX-2) inhibitors may relieve symptoms of anxiety.

COX-2 inhibitors target an enzyme called cyclooxygenase-2, which is produced in many tissues throughout the body and brain. Some COX-2 inhibitors, such as Celebrex, are already widely used to treat arthritis. Studies in people have also suggested these drugs given in combination with antidepressants might also help people with depression, while a recently developed COX-2 inhibitor, called LM-4131, stops the breakdown of chemicals produced in the brain that interact with the same brain receptors that marijuana does. In animal studies, this new drug reduced anxious behaviors. However, scientists didn't know if the drug would also reduce anxious behaviors after animals were exposed to a traumatic or stressful experience.

Now, Gamble-George et al. – including some of the same scientists that created LM-4131 – show this drug does reduce anxious behavior in mice after a traumatic or stressful experience. In the experiments, mice were stressed with an electric shock to their feet and then treated with LM-4131, Celebrex, another COX-2 inhibitor sometimes used in humans, or an inactive control. The mice were then placed in new environments or mazes high off the ground that usually trigger anxious behavior. The mice treated with any of the three COX-2 inhibitors were less likely to act like they were anxious than the mice treated with the control solution.

Gamble-George et al. then showed that LM-4131 changes how cells in the brain become excited in both isolated mouse brain cells and live mice. Short-term stress in humans is known to excite cells in an area of the brain called the amygdala, which controls how we respond to stress or trauma. The experiments showed that LM-4131 decreases this activity and decreases anxiety in stressed mice. Future animal and human studies are needed to confirm that COX-2 inhibitors are a useful treatment for stress disorders, and learn more about how they work.

COX-2 catalyzes synthesis of inflammatory prostaglandins (PGs) from free arachidonic acid and is expressed constitutively and in an activity-dependent manner in neurons (*Kaufmann et al., 1996*; *Kulmacz and Lands, 1984*; *Yamagata et al., 1993*). Based on the immune hyperactivity hypothesis of depression (*Felger and Miller, 2014*; *Haroon et al., 2012*; *Miller et al., 2009*; *Raison and Miller, 2013*), several clinical trials have demonstrated efficacy of COX-2 inhibition as an augmentation strategy to selective-serotonin reuptake inhibitors (SSRIs) for the treatment of major depressive disorder (MDD) (*Akhondzadeh and Jafari, 2010*; *Akhondzadeh et al., 2009*; *Muller et al., 2006*). A number of preliminary studies also suggest that inhibition of COX-2 could have anxiolytic- and antidepressant-like behavioral effects in animal models. Specifically, chronic Celecoxib treatment prevents the development of anhedonia after chronic unpredictable stress (*Guo et al., 2009*), and chronic COX-2 inhibition reduces immobilization stress-induced hypoactivity, memory deficits (*Kumari et al., 2007*), and anxiety-like behavior in the mirror chamber test (*Dhir et al., 2006*).

Interestingly, in addition to generating PGs, COX-2 also metabolizes endogenous cannabinoids (eCBs; anandamide and 2-arachidonoylglycerol) (*Kozak et al., 2000*; *2001*), and several eCB augmentation strategies, including inhibition of fatty acid amide hydrolase (FAAH) and monoacylglycerol lipase (MAGL), have been found to attenuate anxiety- and depression-related behaviors (*Fu et al., 2003*; *Griebel and Holmes, 2013*; *Gunduz-Cinar et al., 2013*; *Hill et al., 2009*; *Ruehle et al., 2012*). However, little is currently understood about the possible contribution of eCB signaling to the behavioral and therapeutic actions of COX-2 inhibitors in mood and anxiety disorders.

We have recently developed a class of COX-2 inhibitors that differentially inhibit COX-2 enzymatic activity based on the substrate: arachidonic acid or eCBs (*Duggan et al., 2011*; *Hermanson et al., 2013*). Specifically, these 'substrate-selective COX-2 inhibitors' (SSCIs) inhibit COX-2 activity when eCBs are used as substrates, without affecting the ability of COX-2 to generate

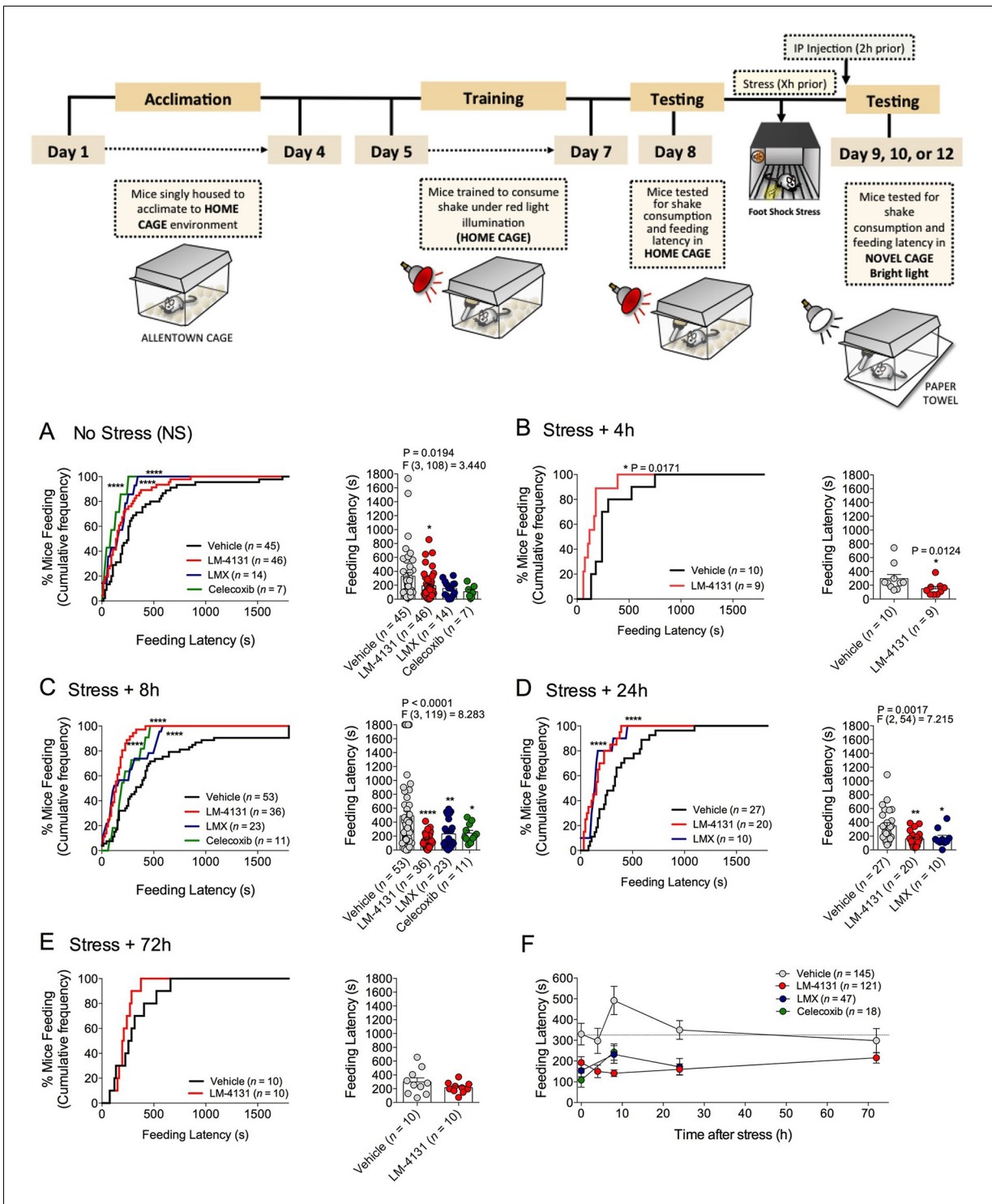

**Figure 1.** Effects of COX-2 inhibition in the NIH assay. Effects of vehicle, LM-4131, Lumiracoxib, and Celecoxib administered 2 hr prior to behavioral testing on feeding latency under non-stressed (control conditions) (**A**), and 4 hr (**B**), 8 hr (**C**), 24 hr (**D**), and 72 hr (**E**) after foot-shock stress (experimental design shown above). Cumulative distribution curves depict percentage of mice feeding at each latency time point (sec), while bar graphs represent mean ± S.E.M feeding latency for each group and individual data points. Each point represents 1 mouse. (**F**) Re-plotted data from A-E in time-course format depicting effects of drugs on feeding latency over time. Significant F and P values from one-way ANOVA noted above bar graphs; *p<0.05, **p<0.01, ****p<0.0001 by Holm-Sidak post hoc multiple comparisons test in bar graphs. For cumulative frequency distributions, *p<0.05, ****p<0.0001 by K-S test.

*Figure 1 continued on next page*

*Figure 1 continued*

The following figure supplements are available for figure 1:

**Figure supplement 1.** Effects of COX-2 inhibition on feeding latency in the NIH assay in 4-month-old male mice.
**Figure supplement 2.** Effects of COX-2 inhibition on feeding latency in the NIH test in female mice.
**Figure supplement 3.** Effects of COX-2 inhibition on feeding latency in the NIH test after sub-chronic stress.
**Figure supplement 4.** Effects of subchronic COX-2 inhibition on feeding latency in the NIH test.
**Figure supplement 5.** Effects of COX-2 inhibition on locomotor activity in the open field test.

PGs (*Hermanson et al., 2013*). The development of these SSCIs provides a powerful tool for dissecting the mechanisms by which pharmacological COX-2 inhibition can modulate a variety of physiological and pathophysiological processes (*Hermanson et al., 2014*).

In the current study, we performed a comparative analysis between the prototypic SSCI, LM-4131, and two traditional COX-2 inhibitors currently or previously approved for pain and arthritis, Lumiracoxib (LMX) and Celecoxib, in a variety of measures of mouse anxiety-, fear- and depression-related behavior. We show that all three COX-2 inhibitors attenuate stress-induced anxiety-like states, with minimal effects on basal anxiety-like, despair-like, and locomotor behaviors. The anxiolytic-like effects of LM-4131 were paralleled by a reduction in anxiety-related BLA neuronal activity ex vivo and in vivo. Collectively, these data support the therapeutic potential of COX-2 inhibition for the treatment of stress-related neuropsychiatric conditions.

## Results

### COX-2 inhibition reduces stress-induced anxiety in the novelty-induced hypophagia assay

We sought to determine the efficacy of pharmacological COX-2 inhibition in preclinical models of affective disorders. Since stress is the major environmental risk factor for the development of mood and anxiety disorders (*Caspi et al., 2003*; *Kessler, 1997*; *McEwen, 2003*), we evaluated the behavioral effects of COX-2 inhibition after stress exposure. The novelty-induced hypophagia (NIH) assay is a well-validated ethologically relevant measure of both anxiety and depressive-related behaviors, encompassing both conflict anxiety and hedonic motivational processes, and is responsive to chronic (but not acute) SSRI treatment (*Cryan and Holmes, 2005*; *Dulawa and Hen, 2005*; *Merali et al., 2003*). We utilized this assay to test the efficacy of a newly developed SSCI, LM-4131, relative to two clinically effective COX-2 inhibitors, LMX and Celecoxib, to reduce basal and stress-induced anxiety-like behavior in mice. Results showed that systemic administration of either LM-4131 or LMX reduced mean feeding latency in (5–7 week old) mice that had been subjected to foot-shock stress 8 or 24, but not 4 or 72, hours earlier (*Figure 1B–E*). Celecoxib, tested at the 8-hr post-stress time point, also reduced mean feeding latency (*Figure 1C*). In contrast, only LM-4131, and not LMX or Celecoxib, produced a small reduction in mean feeding latency in non-stressed controls (*Figure 1A*). It is interesting to note that the anxiety-like effect of foot-shock stress peaked at ∼8 hr after foot-shock, and resolved by 72 hr. The delayed onset of the anxiety-like response to foot-shock may be related to time-dependent neuronal plasticity mechanism induced by the foot-shock exposure. Together, these data show that stress causes an increase in feeding latency between 8 and 24 hr, which was abrogated by the acute SSCI and the traditional COX-2 inhibitor treatment (*Figure 1F*).

Demonstrating the generalizability of these effects across different subjects, stressor durations and drug treatment conditions, we showed that LM-4131, LMX, and Celecoxib decreased mean feeding latency in older, 4-month-old mice, 8 hr after foot-shock stress, but not in non-stressed controls (*Figure 1—figure supplement 1*). Additionally, LM-4131 also reduced mean feeding latency 8-hr post-stress in female mice (*Figure 1—figure supplement 2*). Moreover, to examine whether the

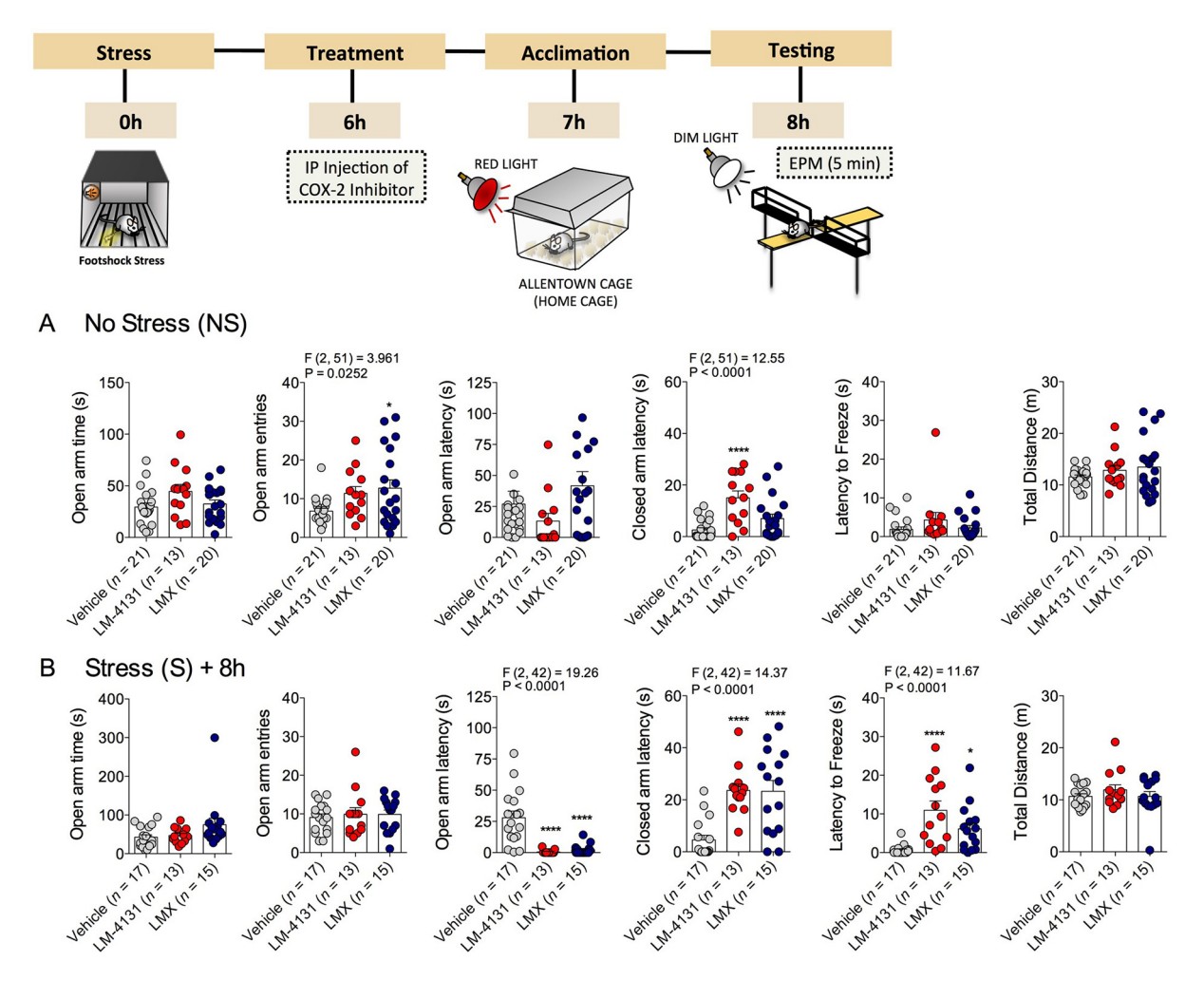

**Figure 2.** Effects of COX-2 inhibition in the elevated-plus maze. (**A**) Effects of LM-4131 (10 mg/kg) and Lumiracoxib (experimental design shown above) under basal non-stressed conditions. (**B**) Effects of LM-4131 and Lumiracoxib 8 hr after acute foot-shock exposure. Significant F and P values for one-way ANOVA shown above bar graphs. *p<0.05, ****p<0.0001 by Holm-Sidak multiple comparisons post hoc test.

The following figure supplement is available for figure 2:

**Figure supplement 1.** Effects of COX-2 inhibition in the EPM in ~4-month-old mice.

SSCI, LM-4131, retained anxiolytic-like efficacy following a more chronic, severe stressor, male mice were exposed to foot-shock stress for 5 consecutive days. Under these conditions, LM-4131 administered 2 hr prior to testing was still able to reduce mean feeding latency in stressed mice, but not non-stressed controls (*Figure 1—figure supplement 3*). Finally, we tested for the effects of repeated daily treatment with LMX (used instead of LM-4131 due to its long half-life (*Mangold et al., 2004*) for 5 consecutive days prior to acute stress exposure. This subchronic treatment regimen reduced feeding latency 8 hr after stress, with no effects in non-stressed controls, thereby mimicking the effects of acute treatment (*Figure 1—figure supplement 4*). There were no effects of this subchronic LMX treatment on body weight in either group (Figure S4). Also of note, neither acute LM-4131 nor LMX treatment affected overall locomotor activity in the open-field assay (*Figure 1—figure supplement 5*), indicating drug-induced reductions in feeding latency were not an artifact of hyperactivity. Taken together, these data demonstrate that COX-2 inhibition exerts a

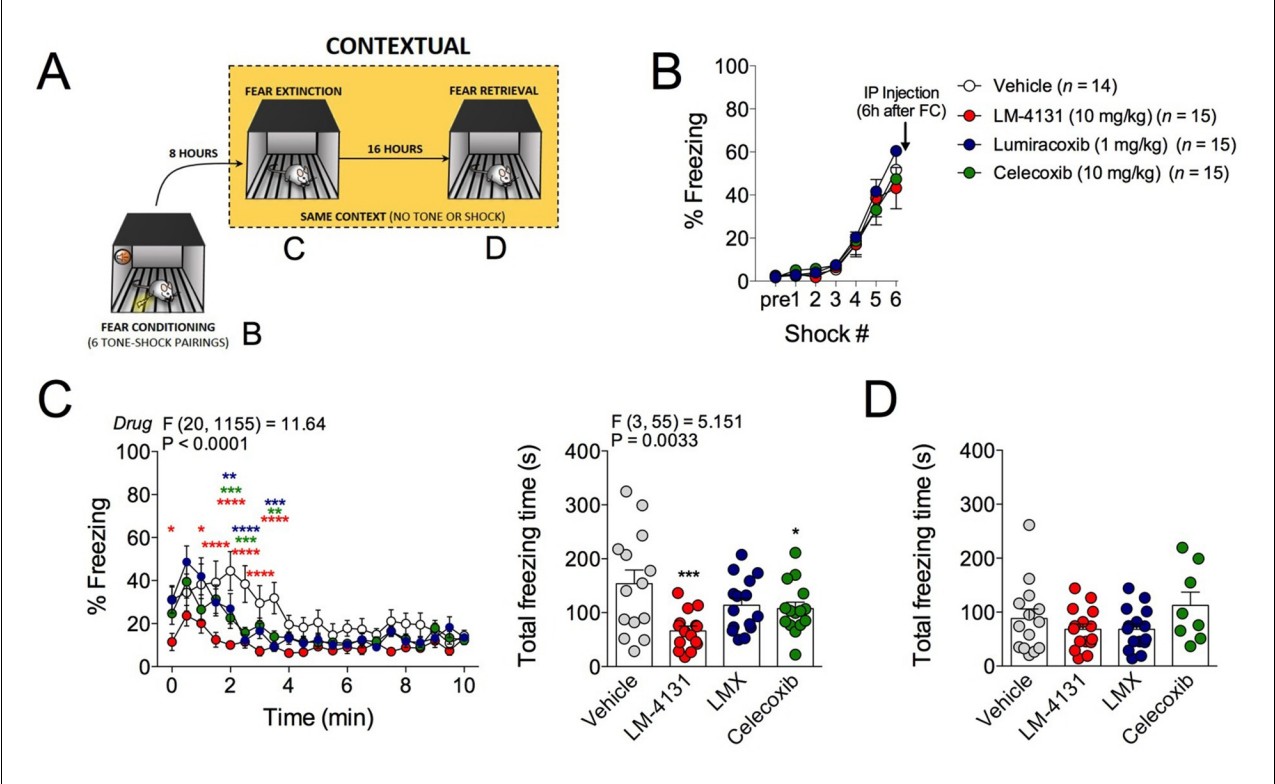

**Figure 3.** Effects of COX-2 inhibition on contextual fear conditioning. (A) Timeline for contextual fear conditioning. Data from each time point shown in correspondingly labeled panel (B–D). (B) No differences in freezing behavior between groups were observed during conditioning. Drugs were administered 6 hr after conditioning (2 hr before fear retrieval). (C) Time course and summary of effects of LM-4131, Lumiracoxib (LMX), or Celecoxib on conditioned fear expression during context re-exposure. (D) Effects of LM-4131, LMX, and Celecoxib on contextual freezing on a second context re-exposure. Significant F and P values for two-way ANOVA (C) and one-way ANOVA (D) shown above figure. *p<0.05, **p<0.01, ***p<0.001, ****p<0.0001 by Holm-Sidak post hoc analysis. Color of * indicates significance level of comparison between each drug and vehicle; red, LM-4131; blue, LMX; green, Celecoxib.

robust, preferential attenuation of stress-induced affective dysregulation across a range of experimental conditions.

## COX-2 inhibition reduces stress-induced anxiety in the elevated-plus maze

We also examined the effects of COX-2 inhibition in a well-established test for anxiety-like behavior, the elevated-plus maze (EPM). In non-stressed mice, drug treatment increased open arm entries and closed arm latency, without affecting other measures (*Figure 2A*). In mice stressed 8 hr earlier, treatment with either LM-4131 or LMX reduced the latency to enter an open arm and increased the latency to enter closed arm and to freeze, but did not alter open arm time or entries or total distance traveled (*Figure 2B*). In a separate cohort of 4-month-old mice, LM-4131 and LMX reduced open-arm latency and latency to freeze regardless of prior stress exposure (*Figure 2—figure supplement 1*). These data demonstrate the effects of COX-2 inhibition are evident across behavioral assays, and are not restricted to appetitive motivated behavioral tests such as the NIH assay.

## COX-2 inhibition reduces contextual fear

In addition to the anxiety-like phenotype caused by foot-shock stress exposure, re-exposure to the foot-shock context induces a conditioned fear response in the form of freezing. We therefore tested the effects of COX-2 inhibition on this conditioned fear response by administering LM-4131, LMX or Celecoxib 6 hr after foot-shock and then testing for contextual freezing 2 hr later (see *Figure 3A* for experimental design). Acquisition of freezing behavior during foot-shock was not different between

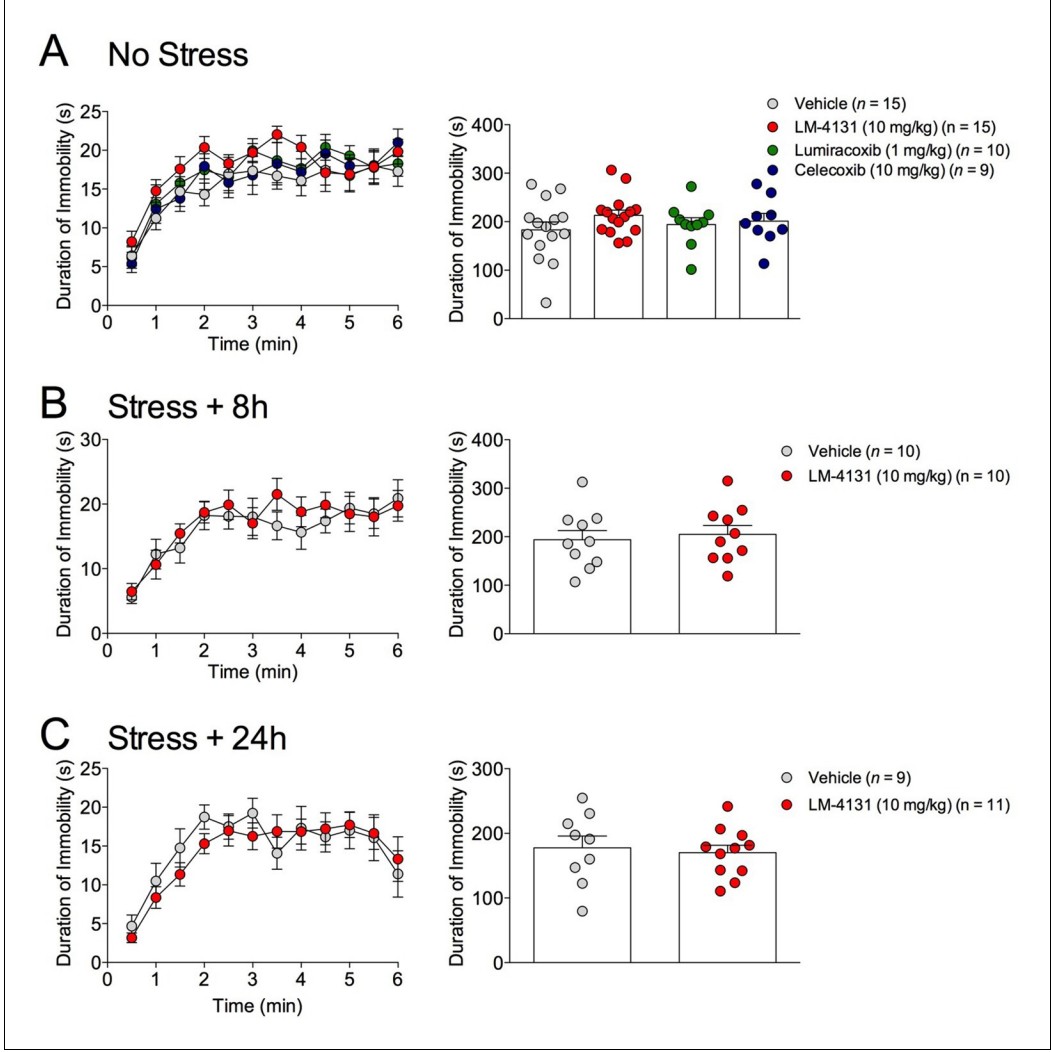

**Figure 4.** Effects of COX-2 inhibition in the TST. (**A**) Effects of LM-4131, LMX, and Celecoxib on immobility duration under non-stressed conditions. (**B–C**) Effects of LM-4131 on immobility duration 8 and 24 hr after foot-shock exposure.

treatment groups prior to drug administration (**Figure 3B**). However, drug treatment reduced expression of contextual fear measured 8 hr post-conditioning (**Figure 3C**). Freezing levels were low on a subsequent drug-free retrieval test 24 hr after conditioning, indicating extinction or decay of the conditioned-fear response, and no residual drug treatment effect was evident at this time point (**Figure 3D**). These data extend our stress-induced anxiety-related findings by demonstrating attenuation in the expression of contextual conditioned fear by COX-2 inhibition.

## COX-2 inhibition does not affect behavioral despair or sucrose preference

The effects of COX-2 inhibition were next tested in two assays sensitive to monoamine antidepressant treatment, the tail suspension test (TST) and the sucrose preference test (**Cryan and Holmes, 2005**). Interestingly, we found that neither LM-4131, LMX, nor Celecoxib altered TST immobility time in non-stressed mice (**Figure 4A**) and that LM-4131 also had no effect on this behavior in mice given foot-shock stress 8 or 24 hr previously (**Figure 4B–C**). In a similar vein, LM-4131 had no effect on sucrose consumption or preference, either under non-stressed conditions or 8 hr following foot-shock (**Figure 5**). These data highlight a degree of specificity in the behavioral effects of acute COX-2 inhibition.

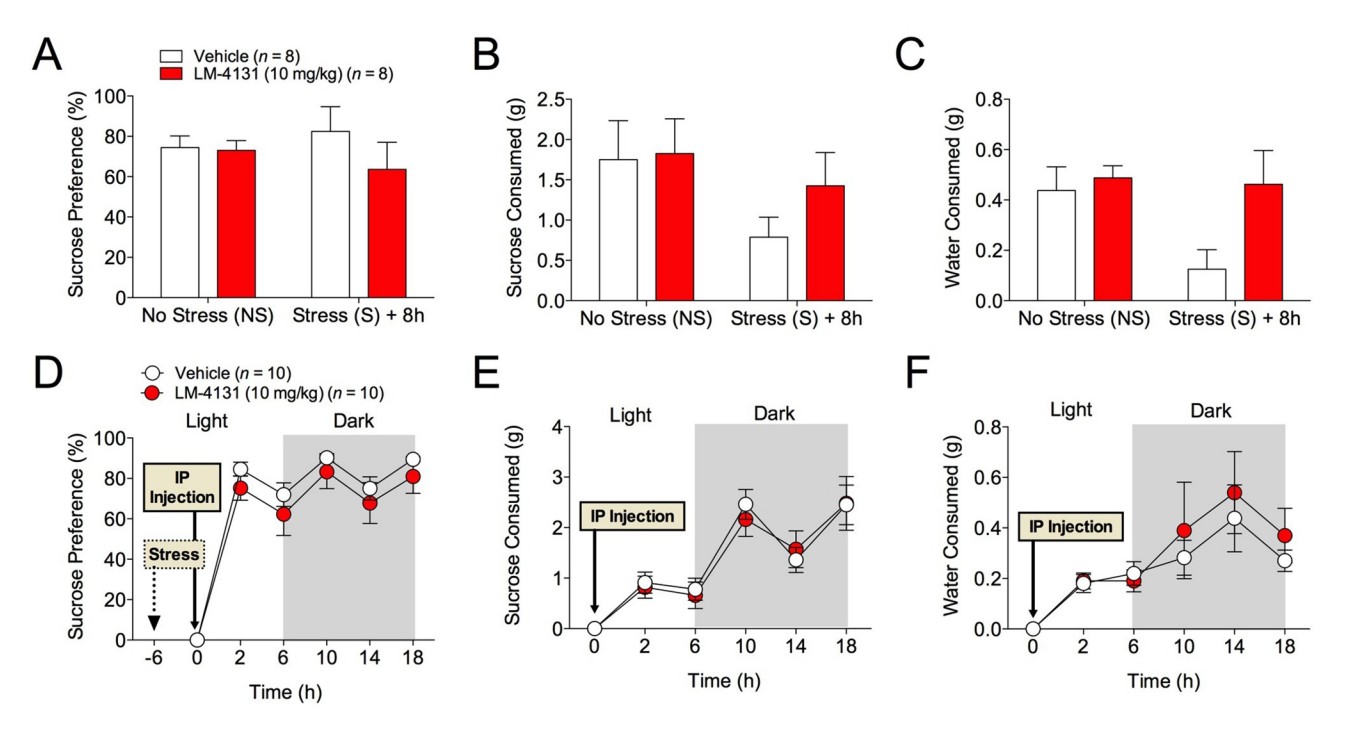

**Figure 5.** Effects of COX-2 inhibition in the sucrose preference test. (**A–C**) 2 hr cumulative sucrose preference, sucrose consumption, and water consumption after vehicle or LM-4131 treatment in non-stressed mice and 8 hr after foot-shock exposure. Testing was conducted during the light phase of the circadian cycle. (**D–F**) Effects of LM-4131 or vehicle on sucrose preference, sucrose consumption, and water consumption during the light and dark phases of the circadian cycle. Mice were treated with LM-4131 or vehicle 8 hr after foot-shock exposure and tested for 16 hr (4 hr during the light cycle and 12 hr during the dark cycle).

## COX-2 inhibition reduces stress-induced anxiety-like behavior via SK channel activity

As we have previously shown that LM-4131 exerts behavioral effects via type-1 cannabinoid receptor (CB1R) activation subsequent to anandamide elevation (*Hermanson et al., 2013*), we tested whether the anxiolytic-like effects of LM-4131 after foot-shock stress exposure were mediated via CB1R signaling. Replicating data shown in *Figure 1A*, under non-stressed control conditions, LM-4131 produced a slight but significant reduction in feeding latency, and this effect was blocked by co-treatment with the CB1R antagonist Rimonabant (2 mg/kg) (*Figure 6—figure supplement 1*). By contrast, LM-4131-induced decreases in feeding latency, that were seen 8 hr after stress, were not prevented by co-administering a low (2 mg/kg) or higher (5 mg/kg) dose of Rimonabant (*Figure 6A–B*). These data show that while the anxiolytic-like effects of LM-4131 are CB1R dependent under non-stressed conditions, in line with previous work (*Hermanson et al., 2013*); however, following stress these effects become CB1R-insensitive and likely mediated by other mechanisms.

Although CB1R activation may not mediate the anxiolytic-like effects of COX-2 inhibition in stressed mice, the eCB anandamide can activate several molecular targets in addition to CB1R, including CB2R, transient receptor potential cation channel, subfamily V, member 1 (TRPV1) receptors, and SK channels (*Di Marzo and De Petrocellis, 2012*; *Ross, 2003*; *Wang et al., 2011*). Moreover, LM-4131 exerts preferential effects on anandamide over 2-AG (*Hermanson et al., 2013*). We therefore tested whether blocking these alternate anandamide targets affected the anxiolytic-like effects of LM-4131 after stress. Neither co-treatment with the CBR2 antagonist, SR144528 (3 mg/kg), nor the TRPV1 antagonist, Capsazepine (10 mg/kg), blocked LM-4131-induced reductions in feeding latency 8 hr after stress (*Figure 6C*). However, co-administration of the SK channel inhibitor, Apamin (0.4 and 0.8 mg/kg), blocked the ability of LM-4131, as well as LMX or Celecoxib, to reduce stress-induced anxiety-like behavior (*Figure 6D*). To further substantiate a role for SK channels in the

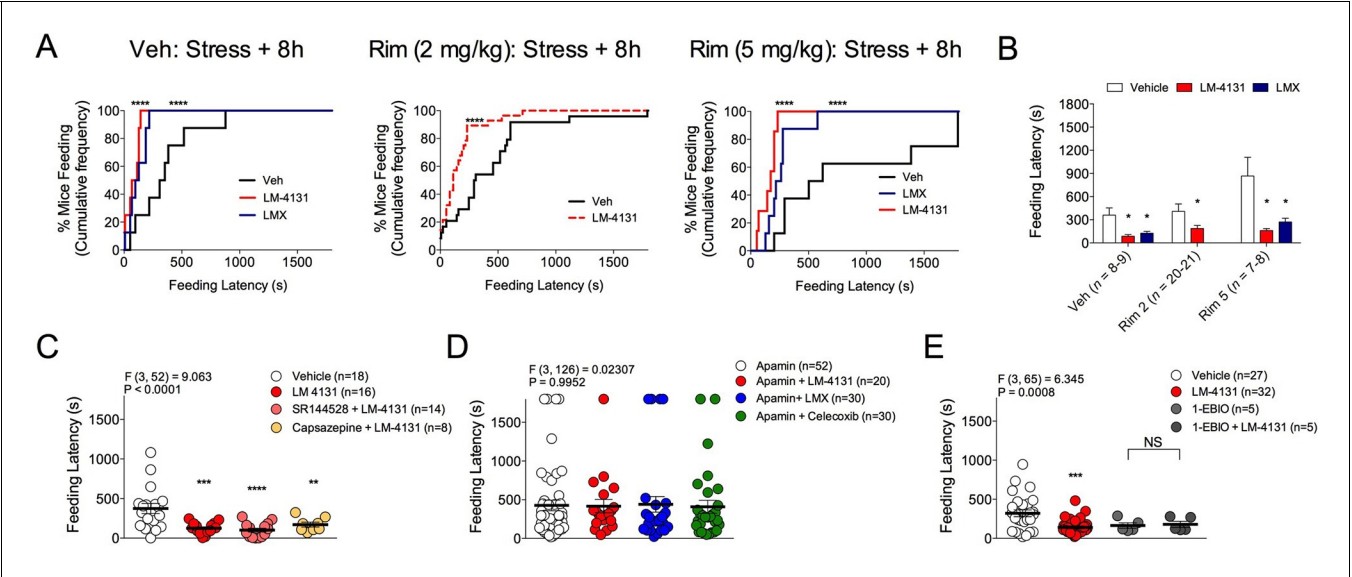

**Figure 6.** Receptor mechanisms mediating anxiolytic-like effects of COX-2 inhibition. (**A**) Cumulative feeding latency distribution curves for vehicle, LM-4131, and Lumiracoxib treated mice in the presence of vehicle, or the CB1R antagonist Rimonabant (2 or 5 mg/kg) co-treatment. (**B**) Mean feeding latency for each group tested 8 hr after stress exposure. (**C**) Effects of the CB2R antagonist, SR144528 (3 mg/kg), or the TRPV1 antagonist, Capsazapine (10 mg/kg), on LM-4131-induced reductions in feeding latency tested 8 hr after stress exposure. (**D**) Effects of the SK channel inhibitor, Apamin (0.4 or 0.8 mg/kg; combined data shown), on LM-4131, LMX, and Celecoxib-induced reductions in feeding latency tested 8 hr after stress exposure. (**E**) Effects of the SK channel activator, 1-EBIO (5 mg/kg), on feeding latency in the presence and absence of LM-4131 tested 8 hr after stress exposure. Significant F and P values from one-way ANOVA noted above bar graphs; *p<0.05, **p<0.01, ***p<0.001, ****p<0.0001 by Holm-Sidak post hoc comparisons test in bar graphs. For cumulative frequency distributions, ****p<0.0001 by K-S test.

The following figure supplements are available for figure 6:

**Figure supplement 1.** Effects of CB1R antagonism on LM-4131-induced reductions in feeding latency in the NIH test.

**Figure supplement 2.** Effects of Apamin and 1-EBIO on locomotor activity.

effects of LM-4131 after foot-shock stress, we tested whether direct activation of SK channels was sufficient to mimic the anxiolytic-like effects of LM-4131. Consistent with earlier data, LM-4131 reduced feeding latency in mice stressed 8 hr earlier (*Figure 6E*). The SK channel activator 1-EBIO also reduced feeding latency, and the combined administration of 1-EBIO and LM-4131 did not produce a further decrease in feeding latency relative to either drug alone (*Figure 6E*). Apamin and 1-EBIO had minimal effect on locomotor activity when administered alone at doses used above (*Figure 6—figure supplement 2*), showing these drug effects on feeding latency were not artifacts of locomotor disturbances.

We have previously shown that eCB signaling effects on food consumption and feeding latency in the assay are dissociable (*Gamble-George et al., 2013*). Therefore, we extended our analyses to measure drug effects on total food consumption in the NIH assay to clarify whether COX-2 inhibition affects feeding via a CB1R-dependent or independent mechanism. We found that reductions in food consumption produced by testing in a novel cage (relative to the home cage) (*Figure 7A*) were partially reversed by LM-4131 in mice with prior foot-shock stress exposure, albeit not in all experiments (*Figure 7B–F*). In addition, examination of feeding behavior in our CB1R antagonist studies indicated that, in contrast to the negative effects of CB1R blockade on feeding latency after stress exposure, the ability of LM-4131 and LMX to normalize stress-induced deficits in food consumption was completely blocked by either 2 or 5 mg/kg Rimonabant (*Figure 7G–H*). Thus there appear to be dissociable contributions of CB1R to the stress-related consummatory and anxiolytic-like effects of COX-2 inhibition (*Gamble-George et al., 2013*).

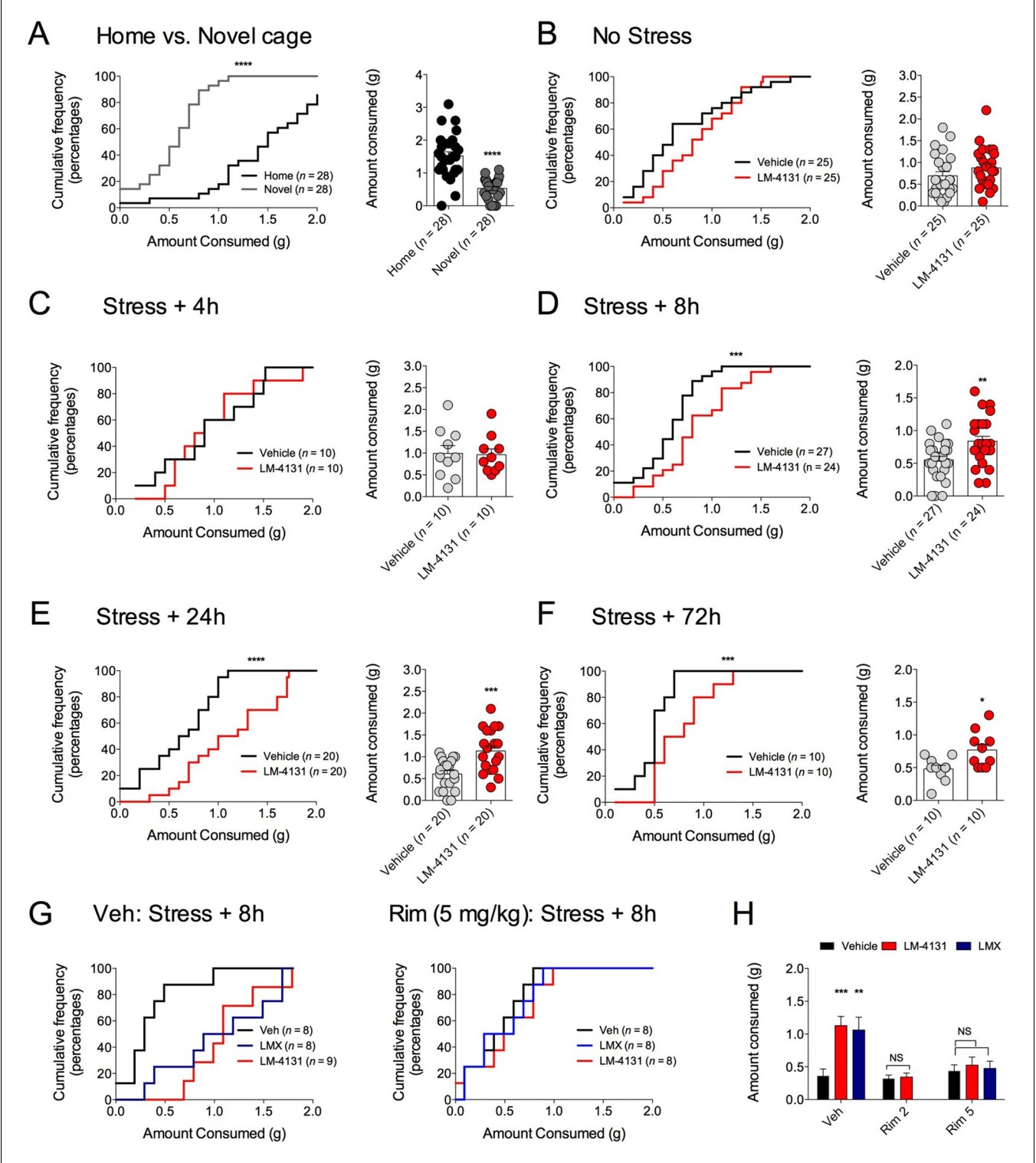

**Figure 7.** Effects of COX-2 inhibition on food consumption in the NIH test. (**A**) Cumulative distribution curves depicting proportion of mice consuming a given amount of food, and mean consumption amounts in the home cage and novel cage setting. (**B–F**) Effects of LM-4131 on Ensure consumption compared to vehicle treatment under control (non-stressed) conditions (**B**), 4 hr (**C**), 8 hr (**D**), 24 hr (**E**), and 72 hr (**F**) after foot-shock stress. (**G–H**) Effects of Vehicle, LM-4131 or Lumiracoxib on consumption in the presence of vehicle or Rimonabant (2 or 5 mg/kg) co-treatment. In bar graphs *p<0.05, **p<0.01, ***p<0.001 by unpaired two-tailed t-test (**B–F**) or by Holm-Sidak multiple comparisons test (**G**). For cumulative frequency distributions, ***p<0.001, ****p<0.0001 by K-S test.

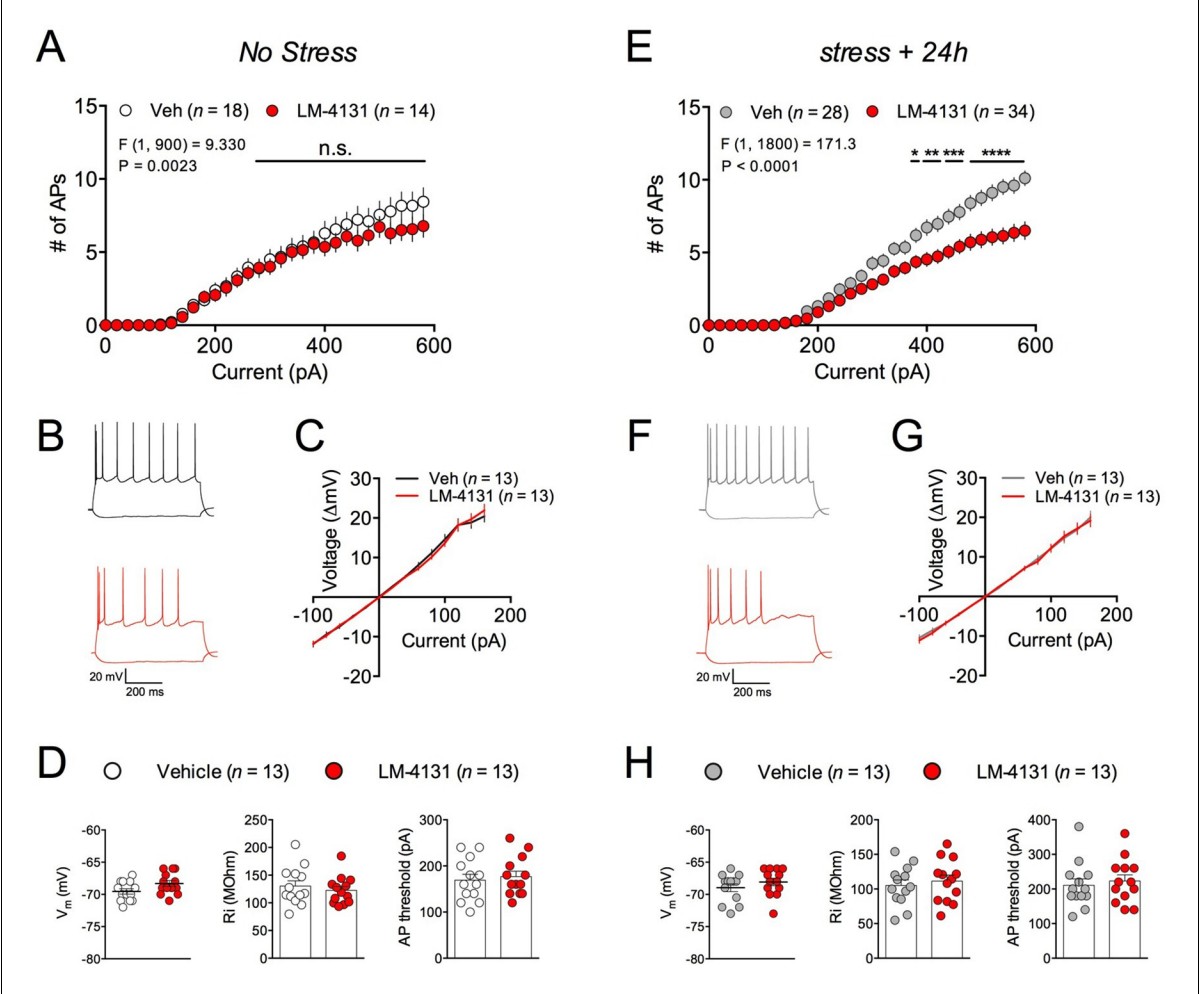

**Figure 8.** Effects of COX-2 inhibition on ex vivo BLA neuronal excitability. (**A–B**) Effects of LM-4131 (10 µM) on BLA neuron excitability, (**C**) I/V relationship, (**D**) resting membrane potential ($V_m$), input resistance ($R_i$), and action potential (AP) threshold in non-stressed mice. (**E–F**) Effects of LM-4131 on BLA neuron excitability, (**G**) I/V relationship, (**H**) resting membrane potential ($V_m$), input resistance ($R_i$), and action potential (AP) threshold in mice 24 hr after foot-shock stress. F and P values for drug effects by two-way ANOVA shown in (A and E), *p<0.05, **p<0.01, ***p<0.001, ****p<0.0001 by post hoc Holm-Sidak test in (**A**) and (**E**).

The following figure supplements are available for figure 8:

**Figure supplement 1.** Effects of Rimonabant, Apamin, and Capsazepine on LM-4131-induced reductions in BLA neuron excitability.

**Figure supplement 2.** Effects of FAAH inhibition on ex vivo BLA neuronal excitability.

## LM-4131 reduces intrinsic excitability of amygdala neurons ex vivo

In order to gain insight into the synaptic and cellular mechanisms by which LM-4131 could affect stress-induced anxiety, we utilized ex vivo whole-cell patch-clamp electrophysiological approaches. Specifically, we determined the effects of LM-4131 on cellular and synaptic physiology in the BLA, given its key role in regulating anxiety and stress-response integration (*Janak and Tye, 2015*; *Pape and Pare, 2010*). In control, non-stressed mice, incubation of brain slices with LM-4131 (10 µM) for 2 hr did not affect intrinsic excitability (*Figure 8A–B*) or current-membrane voltage relationship (*Figure 8C*) of BLA principal neurons, and did not affect resting membrane potential, input resistance, or action potential threshold (*Figure 8D*). Conversely, in brain slices prepared 24 hr after foot-shock stress, LM-4131 incubation resulted in a significant decrease in intrinsic excitability of BLA neurons (*Figure 8E–F*). This reduction was not associated with changes in the current-membrane

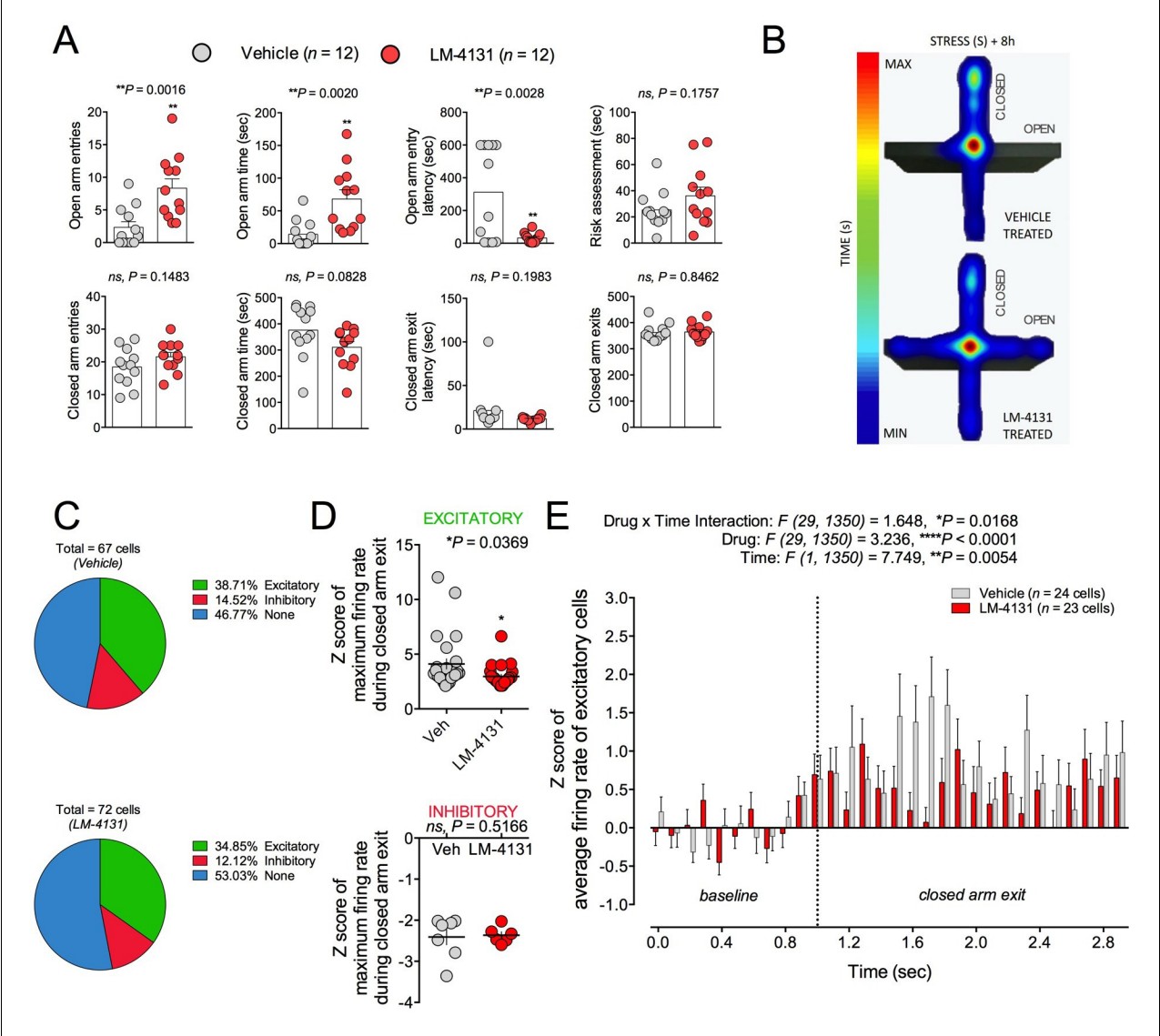

**Figure 9.** Effects of COX-2 inhibition on in vivo activity of BLA neuronal activity. (**A**) LM-4131 decreases latency to enter and increases entries into and time spent in the open arms of the EPM 8 hr following stress exposure. (**B**) Representative heat maps of time spent in the closed (vertical) versus open (horizontal) arms of the EPM. (**C**) Percentage of total cells recorded that were responsive upon exiting a closed arm. (**D**) LM-4131 reduced the maximum firing rate of cells responsive to closed arm exit, but had no effect on cells inhibited by closed-arm exit. (**E**) LM-4131 reduced the increase in average firing rate following closed arm exit in responsive cells. *p<0.05, **p<0.01 by two-tailed unpaired t-test. F and P values for two-way ANOVA shown in (**E**).

The following figure supplement is available for figure 9:

**Figure supplement 1.** Schematic diagram of in vivo electrophysiological recording sites.

voltage relationship (*Figure 8G*), or the resting membrane potential, input resistance, or action potential threshold (*Figure 8H*). These data suggest that one mechanism by which LM-4131 reduced stress-induced behavioral dysregulation is via reductions in the overall activity of BLA neurons via reductions in intrinsic excitability. Consistent with our behavioral data, that the ability of LM-4131 to reduced neuronal excitability after stress exposure was blocked by the SK channel blocker, Apamin, but not the CB1 antagonist SR141716 or the TRPV1 antagonist, Capsazepine (*Figure 8—figure supplement 1A–C*).

Given our behavioral data had indicated a role for COX-2-mediated enhancement of anandamide signaling in the anxiolytic effects of LM-4131, we tested whether directly increasing anandamide levels, via FAAH inhibition, mimicked the effects of LM-4131 on cellular excitability and glutamatergic transmission. Indeed, incubation of brain slices from both control and stressed mice with the FAAH inhibitor, PF-3845 (10 µM), resulted in a decrease in cellular excitability (*Figure 8—figure supplement 2*). These data suggest LM-4131 may reduce stress-induced anxiety-like behavior by dampening neuronal activity in the BLA and, moreover, that these cellular effects can be mimicked by anandamide augmentation via FAAH inhibition.

## LM-4131 dampens anxiety-related amygdala neuronal activity in vivo

Amygdala neuron activity has been linked to the expression of anxiety-like behaviors (*Janak and Tye, 2015*; *Rau et al., 2015*). On this basis and given our ex vivo physiological data, we asked whether LM-4131-induced reductions in BLA neuronal excitability were evident in behaving mice by conducting in vivo single unit recordings. We employed the EPM for this experiment because preliminary work indicated robust behavioral profiles and drug effects under the chronic recording conditions. That is, during recordings, mice treated with LM-4131 and tested 8 hr after stress showed increased open arm time and entries and reduced open arm latency, as compared to vehicle treated controls, consistent with an anxiolytic-like effect in this test (*Figure 9A–B*). It is interesting to note that under these recording conditions mice exhibited high levels of anxiety with open arm time ∼10 s, and LM-4131 was able to increase open arm time and decrease latency to open arm entry. In contrast, in previous studies (*Figure 2*), LM-4131 only affected open arm latency. Given that open arm time in these studies was ∼25–40 s, these data suggest that the open arm latency measure is more sensitive to LM-4131 treatment, but as anxiety levels increase, LM-4131 can affect more conventional measures of anxiety in the EPM including open arm time.

While mice were exploring the EPM, we recorded 62 and 66 single units from vehicle and LM-4131-treated mice, respectively (n=12 mice per group; recording sites depicted in *Figure 9—figure supplement 1*). Firing rates of each unit were analyzed during exit from the safety of the closed arms, and were classified as being excitatory (at least 1 of 10 100-ms time bins following closed arm exit had an average Z-score of >2, relative to a 1 sec pre-exit 'baseline'), inhibitory (converse of excitatory), or unaltered (no significant increase or decrease). The proportion of units falling into each of these 3 categories was no different between vehicle and LM-4131 treated groups (*Figure 9C*). However, the magnitude of the increase in excitatory unit activity on exiting the closed arm was significantly lower in LM-4131 treated mice than vehicle controls (*Figure 9D–E*). Furthermore, the effect of the drug was specific to excitatory activity, with no change evident for units exhibiting inhibitory activity on closed arm exits (*Figure 9D*). The finding that COX-2 inhibition blunted BLA unit excitation in vivo fits well with our ex vivo slice results and, taken together, suggest a neurophysiological correlate of the anxiolytic-like actions of LM-4131.

## Discussion

Here we show that acute treatment with the novel SSCI, LM-4131, as well as two traditional COX-2-selective inhibitors, LMX and Celecoxib, acutely reduces stress-induced behavioral dysregulation in two separate assays (the NIH and EPM), and reduces expression of conditioned fear, without affecting locomotor activity, sucrose preference, or despair-like behaviors. These effects generalized to male mice of different ages, female mice, and were evident even after repeated stress exposure. Moreover, the behavioral effects of COX-2 inhibition were maintained after subchronic treatment indicating a lack of drug tolerance. Together, these results suggest that SSCIs and traditional COX-2 inhibitors could represent viable therapeutic approaches for stress-related neuropsychiatric disorders (*Muller, 2010*).

The three COX-2 inhibitors used in the present study represent different chemical scaffolds: LM-4131 is a morpholinoamide of indomethacin, while LMX is an arylacetic acid, and Celecoxib is a diarylheterocycle. LM-4131 is also a weak antagonist of CB2R (*Gaetani et al., 2003*); however, CB2R activation (*Bahi et al., 2014*; *Garcia-Gutierrez and Manzanares, 2011*), but not inhibition (*Gamble-George et al., 2013*), has been shown to have anxiolytic-like effects, and we have previously shown that CB2R blockade does not affect anxiety-like behavior in the NIH assay (*Gamble-George et al., 2013*). Indeed, we were unable to prevent the anxiolytic-like effects of LM-4131 with a CB2R

antagonist. The similar anxiolytic-like profile of the two other chemically distinct COX-2 inhibitors, which have no known activity at cannabinoid receptors, also argues against a CB2R contribution to the anxiolytic effects of LM-4131.

In principle, COX-2 inhibition could produce anxiolytic and antidepressant actions via one or more mechanisms of action: 1) preventing PG synthesis, 2) increasing eCB levels, and 3) concomitantly reducing levels of prostamides and prostaglandin-glycerol esters (COX-2 eCB oxygenation products). On the one hand, the prior observation that COX-2 inhibition is associated with reduced stress-induced PG levels, cytokine levels, and other markers of neuroinflammation supports the first mechanism (*Dhir et al., 2006*; *Kumari et al., 2007*; *Munhoz et al., 2008*). On the other hand, the current results provide some support for the alternate mechanisms. For example, the SSCI, LM-4131, does not affect production of pro-inflammatory PGs (*Hermanson et al., 2013*), yet is as effective as traditional COX-2 inhibitors in reducing stress-induced anxiety-like behavior. In addition, SK channel blockade and to a lesser extent, CB1R antagonism, both of which are molecular targets of the eCB anandamide, block some behavioral effects of LM-4131, LMX, and Celecoxib. Consistent with our data, systemic and intra-BLA SK channel activation has been shown to reduce stress-induced anxiety and expression of conditioned-fear (*Atchley et al., 2012*; *Rau et al., 2015*). Finally, it is also possible that reductions in prostamides or prostaglandin-glycerols contribute to the anxiolytic-like actions of traditional COX-2 inhibitors and especially SSCIs. However, very little is currently known regarding the physiological role of these metabolites in the regulation of stress responses or behavior in general. Future studies will thus be aimed at determining the differential contributions of these three potential mechanisms of action of COX-2 inhibitors to reduce stress-related pathology.

At a cellular level, LM-4131 decreased amygdala neuron excitability in stressed, but not control, mice. This effect was mimicked by the FAAH inhibitor, PF-3843, which causes robust increases in anandamide levels (*Lee et al., 2015*). Taken together, these findings support the notion that LM-4131 affects amygdala cellular physiology indirectly via enhancing anandamide signaling. We also found the decrease in BLA neuron excitability induced by LM-4131 ex vivo was blocked by the SK channel antagonist Apamin, but not the CB1 or TRPV1 antagonists, mirroring our behavioral studies. It is unlikely that that the cellular effects of LM-4131 are mediated via decreases in PG signaling since our in vitro data indicate that LM-4131 is a very weak inhibitor of AA oxygenation by COX-2 (*Hermanson et al., 2013*). However, it is remains possible that the neurophysiological effects of LM-4131 are mediated in part via inhibition of prostamide signaling; a hypothesis that will require further testing.

We also investigated the in vivo neurophysiological correlates of LM-4131-induced anxiolytic-like effects. Consistent with our ex vivo electrophysiological results, in vivo single-unit recordings revealed that the anxiolytic effects of LM-4131 were associated with a significantly attenuated increase in BLA neuronal excitability that was evident when mice left a relatively safe location. Given the strong link between amygdala neuron activity and generation of anxiety and fear states (*Janak and Tye, 2015*), these data suggest that LM-4131 could reduce anxiety-like behavior by modulating encoding of negative valence by BLA neurons; though further studies will be required to substantiate this idea. Similarly, whether LM-4131 induced AEA augmentation, and subsequent SK channel activation, contributes to the reduction in BLA neuron activity in vivo remains to be determined.

Several clinical trials and meta-analyses of trials have now confirmed the efficacy of COX-2 inhibition as an effective augmentation strategy to SSRIs in MDD (*Abdallah et al., 2015*; *Akhondzadeh and Jafari, 2010*; *Akhondzadeh et al., 2009*; *Faridhosseini et al., 2014*; *Fond et al., 2014*; *Muller et al., 2006*). In addition, COX-2 mRNA is increased in the circulation of patients with MDD (*Galecki et al., 2012*). These findings are generally consistent with the known role of COX-2 as an immediate-early gene up-regulated by neurons in response to various types of stress (*Kaufmann et al., 1996*; *Kulmacz and Lands, 1984*; *Yamagata et al., 1993*). On this basis, it has been proposed that stress exposure could increase susceptibility to depression via generation of inflammatory PGs generated by COX-2; a contention in line with the neuroimmune hypothesis of depression (*Felger and Miller, 2014*; *Haroon et al., 2012*; *Miller et al., 2009*; *Raison and Miller, 2013*) and the aforementioned clinical efficacy of COX-2 inhibition in MDD.

The current study extends this model by supporting additional mechanisms by which COX-2 could exert therapeutic effects; specifically, via eCB augmentation and possibly reductions in COX-2 eCB oxidative metabolites, which are known to have actions opposing eCBs in some systems

(*Ligresti et al., 2014*; *Sang et al., 2006*). This alternate hypothesis is consistent with data suggesting eCB deficiencies could contribute to the pathogenesis of some anxiety disorders, including PTSD (*Hill and Patel, 2013*; *Jenniches et al., 2015*; *Shonesy et al., 2014*). Our preclinical data support robust effects of COX-2 inhibition in fear and anxiety-domain behaviors, but preliminary analysis of anhedonia and despair-like behavior were negative. Therefore, although anxiety symptoms are highly reported in patients with MDD (*Fava et al., 2006*; *Fried et al., 2015*), and anxiety disorders and MDD are highly co-morbid (*Kessler et al., 2005*), our data suggest clinical trials of COX-2 inhibitors in anxiety and trauma- and stressor-related disorders may also be warranted, and that SSCIs could represent a potentially attractive class of novel therapeutics for mood and anxiety disorders.

## Materials and methods

### Subjects

Male or female ICR (CD-1) mice between 5–7 weeks of age or ~4 months of age were used for all experiments (Harlan, Indianapolis, IN, USA). All mice were housed on a 12:12 light-dark cycle (lights on at 6:00 a.m.) with food and water available *ad libitum*. All studies were carried out in accordance with the National Institute of Health Guide for the Care and Use of Laboratory Animals, and approved by the Vanderbilt University Institutional Animal Care and Use Committee (#M10/227) and by the NIAAA Animal Care and Use Committee (#LBGN-AH-41). All behavioral testing was performed during the inactive light phase of the mouse circadian cycle (between 6:00 am and 6:00 pm).

### Drugs and treatment

The drugs used were the COX-2 inhibitors, LM-4131 (10 mg/kg), Lumiracoxib (1 mg/kg), and Celecoxib (10 mg/kg), the SK channel antagonist, Apamin (0.4 and 0.8 mg/kg), the SK channel agonist, 1-EBIO (5 mg/kg), the CB1R antagonist, Rimonabant (2 and 5 mg/kg), the CB2R antagonist, SR144528 (3 mg/kg), and the TRPV1 antagonist, Capsazepine (10 mg/kg). Rimonabant and SR144528 were gifts from the National Institute of Mental Health Drug Supply Program. LM-4131 was synthesized as previously described (*Hermanson et al., 2013*). Lumiracoxib was obtained from Selleck Chemicals (Houston, TX, USA). Celecoxib and Capsazepine were from Cayman Chemical (Ann Arbor, MI, USA). Apamin and 1-EBIO were from Tocris (Minneapolis, MN, USA). Drugs or vehicle were administered by intraperitoneal injection at a volume of 1 mL/kg in DMSO. Drug pretreatment times were two hours prior to behavioral testing.

### Stress exposure

Mice were moved to the test room for 1 hr habituation and placed in a 30.5 x 24.1 x 21.0 cm (175–177 lux) chamber, which was cleaned in between testing with a chlorine dioxide solution (Vimoba). After a 30-second habituation period, there were 6 x 0.7 mA foot-shocks separated by a 30-second interval, with each shock preceded by a 30-s tone (80 dB, 3000 Hz). Mice were then returned to their home cages and a holding room until further behavioral testing. Detailed methods for all behavioral tests are provided in the Supplemental Materials and methods section.

### Novelty-induced hypophagia (NIH) test

NIH testing was carried out exactly as described previously (*Bluett et al., 2014*; *Gamble-George et al., 2013*). Individually housed mice were habituated to a novel, palatable food (Ensure Homemade Vanilla Shake) in their home cages for 30 min per day for 3 days under red light conditions (40 lux). The day after the final habituation session, mice were presented with the shake in their home cages and the latency to begin feeding and the amount consumed was measured. Mice were then exposed to foot-shock stress as described above and tested in the novel cage at 4, 8, 24, or 72 hr after stress exposure depending on the experiment. Novel cage testing was conducted in a new cage, without bedding, and with a white paper placed under the cage using bright ambient lighting conditions (295 lux). Latency to feed in the novel cage and the amount of Ensure® consumed are reported. Drugs were administered 2 hr prior to novel cage testing.

## Open-field test

Open-field locomotor activity was analyzed and performed as previously described (*Gamble-George et al., 2013*).

## Elevated-plus maze (EPM) test

The apparatus was elevated 47 cm from the ground, and consisted of 2 open arms (30 × 5 cm; 90 lux) and 2 closed arms (30 × 5 × 15 cm; 20 lux) extending from a 5 × 5 cm central area (San Diego Instruments, San Diego, CA, USA). The walls and floor were made from black ABS (acrylonitrile buta-diene styrene) plastic. To begin the test, each mouse was placed in the center, facing an open arm. Time spent, latency to enter, and entries into the open and closed arms, latency to closed arm exit, latency to freeze, and total distance traveled were recorded using CinePlex Behavioral Research System (Plexon Inc, Dallas, TX, USA) for chronic in vivo BLA unit recordings, or, for all other experiments, Any Maze tracking software. The mouse was considered to be in an arm when all four paws were in the arm. The test was 5 min in duration, with the exception of the chronic in vivo BLA unit recording experiment, which was 10 min in length.

## Contextual fear conditioning

Mice were conditioned using the stress exposure procedure described above. Six hours later, mice were administered either vehicle or LM-4131 (10 mg/kg). Two hours later and again 16 hr later, mice were returned to the conditioned chamber for a 10-minute test. The VideoFreeze system (Med Associates) was used to measure freezing: defined as no movement other than breathing and measured when mouse movement fell below a preset motion threshold of 18.

## Tail suspension test (TST)

The TST was performed as previously described (*Gamble-George et al., 2013*).

## Sucrose preference test

Each mouse was given twenty-four hours to choose freely between a 2% sucrose solution and tap water in 50 mL clear plastic conical tubes. To prevent side preference bias, the position of the two tubes was switched at every measurement of the amount of fluid consumed (e.g., 2–4 hr). No previous water or food deprivation was applied before testing. The consumption of the 2% sucrose solution or water was measured by weighing the conical tubes containing the fluids. Percentage sucrose preference was calculated as the ratio of 2% sucrose consumed over total fluid consumption multiplied by 100.

## Chronic in vivo unit recordings

Mice were anesthetized with 2% Isoflurane (Baxter Healthcare, Deerfield, IL, USA) and implanted with 2 × 8 electrode (35-µm) tungsten micro-arrays (Innovative Neurophysiology Inc, Durham, NC, USA) bilaterally targeted at the BLA (3.1 mm lateral, 1.2–2.2 mm posterior, 4.5 mm ventral relative to bregma). Following surgery, mice were individually housed and allowed at least 1 week to recover prior to behavioral testing. Animals were exposed to stress as described above. Five hours after stress exposure, mice received a pre-drug baseline recording session in its home cage. Individual units were identified and recorded for 5 min using the Plexon Omniplex Neural Data Acquisition System. One hour later, mice were administered either LM-4131 (10 mg/kg) or vehicle and, 2 hr later, received a 5-minute post-drug recording session in the home cage. Immediately following this test, mice were assessed on the EPM as described above. Neural data was sorted using the Plexon Offline Sorter and analyzed using NeuroExplorer 5.0 (Nex Technologies, Madison, AL, USA).

For analysis of unit recording data, waveforms were isolated manually using principal component analysis (Offline Sorter, Plexon Inc). To be included in the analyses, spikes had to exhibit a refractory period of at least 1 millisecond. Autocorrelograms from simultaneously recorded units were examined to ensure that no cell was counted twice. Each unit's activity was sorted into 100 millisecond bins, time-locked to exits from the safety of the closed arm and z-scored to average activity during a 1-s pre-exit baseline. Units were classified as either 1) excitatory, if 1 or more post-exit bins during 1-s post-exit exceeded 2 z-scores, 2) inhibitory, if 1 or more post-exit bins was less than 2 z-scores, or 3) non-responsive, if no post-exit bin was > or < 2 z-scores. Analysis of arm entry responsiveness

included 67 cells in vehicle-treated animals, and 72 cells in LM-4131-treated animals. Data reported for raw firing rates included only putative principal neurons; any unit with a baseline firing rate over 7 Hz was excluded (5 vehicle and 3 LM-4131; see [*Likhtik et al., 2006*]).

## Ex vivo slice electrophysiological recordings
Brain slice preparation and whole-cell patch clamp recordings were performed exactly as described previously (*Ramikie et al., 2014*; *Shonesy et al., 2014*; *Sumislawski et al., 2011*).

## Statistics
Statistical analysis of single unit data is above. For all other experiments, statistical significance was calculated by two-tailed unpaired *t*-test or Kolmogorov-Smirnov test (only for frequency distribution plots), or one-way or two-way ANOVA with post hoc Holm-Sidak's multiple comparisons test as noted in figure legends. All statistical analyses were conducted using Prism Graphpad 6 (San Diego, CA, USA). No a priori power analysis was conducted; however, minimum sample sized were based on our published experience with behavioral and physiological assays used. All experiments were performed in 2 or more independent cohorts of animals except in vivo electrophysiological studies. For behavioral studies, all replicates (*n* values) represent biological replicates defined as data derived from a single mouse. Data are presented as mean ± S.E.M. unless otherwise stated in the figure legends. P<0.05 was considered significant throughout. Grubb's test was used to remove outliers.

## Acknowledgements
These studies were supported by NIH grant R01MH100096 to SP, R01GM15431 to LJM, and the NIAAA Intramural Research program (A Holmes). JGG was supported in part by the UNCF/Merck Graduate Science Research Dissertation Fellowship and the Southern Regional Educational Board (SREB)-State Doctoral Scholars Program Doctoral Award. Part of the research in this publication was also supported by the Neurosciences Core of the Vanderbilt Kennedy Center, funded by the EKS NICHD of the NIH under Award U54HD083211. Behavioral studies were carried out at the Vanderbilt Neurobehavioral Core Facility. The content is solely the responsibility of the authors and does not necessarily represent the official views of the NIH.

## Additional information

### Competing interests
LJM: LJM is a co-inventor on a patent #US20150183737 entitled "Composition and method of Substrate-Selective COX-2 inhibition". LJM has a collaborative research contract with Lundbeck Pharmaceuticals. SP: SP is a co-inventor on a patent #US20150183737 entitled "Composition and method of Substrate-Selective COX-2 inhibition". SP has a collaborative research contract with Lundbeck Pharmaceuticals. The other authors declare that no competing interests exist.

### Funding

| Funder | Grant reference number | Author |
| --- | --- | --- |
| National Institute of Mental Health | MH100096 | Sachin Patel |
| National Institute of General Medical Sciences | GM15431 | Lawrence Marnett |

The funders had no role in study design, data collection and interpretation, or the decision to submit the work for publication.

### Author contributions
JCG-G, RB, LH, AK, SP, Conception and design, Acquisition of data, Analysis and interpretation of data, Drafting or revising the article; NH, CGS, HR, AHa, Acquisition of data, Analysis and interpretation of data, Drafting or revising the article; LJM, Conception and design, Drafting or revising the article, Contributed unpublished essential data or reagents; AHo, Conception and design, Analysis and interpretation of data, Drafting or revising the article

## Author ORCIDs

Joyonna Carrie Gamble-George, http://orcid.org/0000-0002-5492-9216
Sachin Patel, http://orcid.org/0000-0001-8052-520X

## Ethics

Animal experimentation: All studies were carried out in accordance with the National Institute of Health Guide for the Care and Use of Laboratory Animals, and approved by the Vanderbilt University Institutional Animal Care and Use Committee (protocol # M10/227) and by the NIAAA Animal Care and Use Committee (protocol #LBGN-AH-41).

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
