## [Decision Letter]

Thank you for submitting your work entitled "Cyclooxygenase-2 inhibition reduces stress-induced affective pathology" for consideration by *eLife*. Your article has been favorably evaluated by a Senior editor and two reviewers, one of whom is a member of our Board of Reviewing Editors.

The reviewers have discussed the reviews with one another and the Reviewing Editor has drafted this decision to help you prepare a revised submission.

The authors of this paper describe the effects of a new class of COX-2 inhibitor that has selective effects on cannabinoid metabolism (degradation) rather than prostaglandin synthesis and which appears to decrease anxiety in several tests conducted in mice. The behavioral analysis is extensive and nicely presented so that readers can get a good feel for the variability and significance of the results. The authors go on to show that the effect on anxiety is likely mediated by increasing cannabinoid signaling and provide electrophysiological evidence indicating that this might be due to a suppressing activity of principle neurons in the BLA via SK channels. Overall, this is a fairly thorough analysis and the authors make a compelling case that this drug could be useful therapeutically.

Essential revisions:

There are two major concerns that require attention. One is that the data from human tissue is not very compelling and would require considerable more experimentation to validate their significance. We suggest that the authors either remove those data or provide validation. The other concerns of reviewer #1 should be addressed, but do not require further experimentation. Second, the authors should determine whether SK channels respond selectively to the drugs as requested by the two comments from reviewer #2.

*Reviewer #1:*

The authors of this paper describe the effects of a new class of COX-2 inhibitor that has selective effects on cannabinoid metabolism (degradation) rather than prostaglandin synthesis and which appears to decrease anxiety in several tests conducted in mice. The behavioral analysis is extensive and nicely presented so that readers can get a good feel for the variability and significance of the results. The authors go on to show that the effect on anxiety is likely mediated by increasing cannabinoid signaling and provide ephys evidence indicating that this might be due to a suppressing activity of principle neurons in the BLA. Overall, this is a fairly thorough analysis and the authors make a compelling case that this drug could be useful therapeutically. Nevertheless, there are a few places where clarification would help and the insertion of human data seemed more distracting than useful.

1) In Figure 1, the authors compare the inducibility of human PTDGS2 mRNA in cultured fibroblasts from controls and depressed patients. IL6 was able to induce mRNA but not IL-1 or TNF. What does induction by one class of inflammatory agents relative to the others mean? Are fibroblasts a relevant indicator of PTDG2 induction in CNS? What mechanism(s) would make PTDG2 more inducible in depressed patients? In the Discussion, the authors say that other studies reveal more baseline PTDGS2 mRNA in depressed patients, but the way the data are presented it is difficult to ascertain the baseline values for these two groups. A plot showing relative mRNA level +/- addition of inducers would be easier to comprehend. Finally, COX-2 enzyme activity is the relevant measure rather than mRNA level or inducibility. Activity should be measured along with effects of the inhibitors on that activity. Overall, I think that the analysis shown in Figure 1 raises more questions than it answers and thus detracts from the rest of the paper.

2) In Figure 2 and Figure 8 the authors examine the effects of stress (foot shock) on latency to consume food and on food consumption in either home cage or novel cage. It is not clear whether the "no stress condition" (panel B in both cases) is in home or novel cage. I assume that it is novel cage. In both cases there is little effect of LM-4131 at 4 hr or 72 hr but there is an effect at other times in between. It would be useful if authors provided some explanation for why it takes more than 4 hrs after foot shock for the drug to become effective and why it dissipates by 72 hr.

3) In the EPM experiments, it is usually the case that latency to enter open arm and total time spend in open arm are both affected. In this case, only the latency is affected, suggesting that the mice get over their anxiety once they enter the open arm the first time. That is quite unusual and deserves some discussion.

4) In Figure 4 think it would be useful to sum the time spent freezing for only the first 4 min rather than the entire 10 min test as freezing has essentially stopped by 4 min.

*Reviewer #2:*

This research provides exciting results that LM-4131 produces selective anxiolytic behavioral responses that were blocked by SK channel inhibitor, but not by CB1R, CB2R or TRPV1R antagonist. Therefore, I feel that this research is suitable for the publication in *eLife*. However, I have two minor concerns. First, do LMX and Celecoxib also produce anxiolytic effects through SK channel? Second, can LM-4131-elicited electrophysiological effects on brain slices and in vivo animals be blocked by SK channel inhibitor, but not by CB1R or TRPV1R antagonist?

---

## [Author Response]

Essential revisions:

There are two major concerns that require attention. One is that the data from human tissue is not very compelling and would require considerable more experimentation to validate their significance. We suggest that the authors either remove those data or provide validation. The other concerns of reviewer #1 should be addressed, but do not require further experimentation. Second, the authors should determine whether SK channels respond selectively to the drugs as requested by the two comments from reviewer #2.

Reviewer #1:

The authors of this paper describe the effects of a new class of COX-2 inhibitor that has selective effects on cannabinoid metabolism (degradation) rather than prostaglandin synthesis and which appears to decrease anxiety in several tests conducted in mice. The behavioral analysis is extensive and nicely presented so that readers can get a good feel for the variability and significance of the results. The authors go on to show that the effect on anxiety is likely mediated by increasing cannabinoid signaling and provide ephys evidence indicating that this might be due to a suppressing activity of principle neurons in the BLA. Overall, this is a fairly thorough analysis and the authors make a compelling case that this drug could be useful therapeutically. Nevertheless, there are a few places where clarification would help and the insertion of human data seemed more distracting than useful.

1) In Figure 1, the authors compare the inducibility of human PTDGS2 mRNA in cultured fibroblasts from controls and depressed patients. IL6 was able to induce mRNA but not IL-1 or TNF. What does induction by one class of inflammatory agents relative to the others mean? Are fibroblasts a relevant indicator of PTDG2 induction in CNS? What mechanism(s) would make PTDG2 more inducible in depressed patients? In the Discussion, the authors say that other studies reveal more baseline PTDGS2 mRNA in depressed patients, but the way the data are presented it is difficult to ascertain the baseline values for these two groups. A plot showing relative mRNA level +/- addition of inducers would be easier to comprehend. Finally, COX-2 enzyme activity is the relevant measure rather than mRNA level or inducibility. Activity should be measured along with effects of the inhibitors on that activity. Overall, I think that the analysis shown in Figure 1 raises more questions than it answers and thus detracts from the rest of the paper.

These data have been removed for the paper.

2) In Figure 2 and Figure 8 the authors examine the effects of stress (foot shock) on latency to consume food and on food consumption in either home cage or novel cage. It is not clear whether the "no stress condition" (panel B in both cases) is in home or novel cage. I assume that it is novel cage. In both cases there is little effect of LM4131 at 4 hr or 72 hr but there is an effect at other times in between. It would be useful if authors provided some explanation for why it takes more than 4 hrs after foot shock for the drug to become effective and why it dissipates by 72 hr.

Indeed, all measurements are in the novel cage, and it is interesting that it takes several hours to see the anxiety-like effect of stress, although we have not examined mice immediately after the foot-shock. We speculate that synaptic plasticity mechanisms triggered by the stress experience require some time to become established, and then normalize after 72 hours. We have added this hypothesis to the text (subsection “LM-4131 reduces intrinsic excitability of amygdala neurons ex vivo”).

3) In the EPM experiments, it is usually the case that latency to enter open arm and total time spend in open arm are both affected. In this case, only the latency is affected, suggesting that the mice get over their anxiety once they enter the open arm the first time. That is quite unusual and deserves some discussion.

It is noteworthy that in the in vivo electrophysiological studies, open arm time is also increased, in addition to latency. In these experiments open arm time is very low due to the anxiogenic conditions of being tethered and having electrophysiological headpieces attached to the mouse. We hypothesize that the latency measure is the most sensitive to COX-2 inhibition, but that under very high anxiety states (such as seen in Figure 9), open arm time can also be affected. We have noted this hypothesis in the subsection “LM-4131 dampens anxiety-related amygdala neuronal activity in vivo“.

4) In Figure 4 think it would be useful to sum the time spent freezing for only the first 4 min rather than the entire 10 min test as freezing has essentially stopped by 4 min.

We have re-analyzed the data in 4D to include only the first 4 minutes, however, the effects are less substantial, indicating that even though freezing is quite low after 4 minutes, it is in fact still lower in the drug treated groups relative to vehicle during the last 6 minutes. Therefore, we prefer to keep the summary data analysis to the entire duration of the experiment.

Reviewer #2:

This research provides exciting results that LM-4131 produces selective anxiolytic behavioral responses that were blocked by SK channel inhibitor, but not by CB1R, CB2R or TRPV1R antagonist. Therefore, I feel that this research is suitable for the publication in eLife. However, I have two minor concerns. First, do LMX and Celecoxib also produce anxiolytic effects through SK channel? Second, can LM-4131-elicited electrophysiological effects on brain slices and in vivo animals be blocked by SK channel inhibitor, but not by CB1R or TRPV1R antagonist?

In addressing the reviewer’s first point, we have added data showing that the excitability reducing effect of LM-4131 is present upon CB1 and TRPV1 blockade, but absent upon incubation with SK channel inhibitor Apamin (Figure 8—figure supplement 1). Additionally, we show that anxiolytic effects of LMX and Celecoxib are also blocked by Apamin (Figure 6), similarly to LM-4131.

While we agree the second point will be interesting to pursue, due to the low throughout in vivo electrophysiological experiments, it was simply not feasible to repeat all the slice work in vivo within the 2-month time frame. However, we have now indicated in the Discussion that the question of whether the same mechanisms which are also operative in the brain slice are operative in vivo, will require further investigation to be conclusive.

Lastly, please note we have removed the sEPSC data form the manuscript for several reasons. Upon repeating these experiments to determine the receptor mechanisms subserving these effects, the results were difficult to interpret. Specifically, it appeared that LM-4131 was not able to reduce sEPSC frequency in the presence of Apamin, Rimonabant, or Capsazepine. However, close inspection of the data revealed that in some cases basal eEPSC frequency was quite low, which could prevent observation of the LM-4131 effect due to floor effects. We feel we need to more thoroughly investigate the reproducibility and mechanistic basis for the sEPSC effect of LM-4131; however, this investigation may go beyond the scope and time limitations of the current revision. We also feel the neuronal excitability data are quite robust (Figure 8–Figure 9 and supplements), and clearer in terms of its mechanistic basis – providing a compelling cellular correlate of our behavioral findings even if the sEPSC data are not included.